# Insurance coverage, stage at diagnosis, and time to treatment following dependent coverage and Medicaid expansion for men with testicular cancer

**Adam B. Weiner** [1], **Stephen Jan**[2], **Ketan Jain-Poster**[1], **Oliver S. Ko**[1], **Anuj S. Desai**[1], **Shilajit D. Kundu**[1]*

1 Department of Urology, Northwestern University Feinberg School of Medicine, Chicago, IL, United States of America, 2 University of Maryland School of Medicine, Baltimore, MD, United States of America

* Shilajit.Kundu@nm.org

## Abstract

**Data Availability Statement:** The authors do not have the right share the third-party data used in this study. The data underlying this study are

### Introduction

We sought to assess the impact of Affordable Care Act Dependent Care Expansion (ACA-DCE), which allowed dependent coverage for adults aged 19–25, and Medicaid expansion on outcomes for men with testicular cancer.

### Methods

Using a US-based cancer registry, we performed adjusted difference-in-difference (DID) analyses comparing outcomes between men aged 19–25 (n = 8,026) and 26–64 (n = 33,303) pre- (2007–2009) and post-ACA-DCE (2011–2016) and between men in states that expanded Medicaid (n = 2,296) to men in those that did not (n = 2,265)pre- (2011–2013) and post-Medicaid expansion (2015–2016).

### Results

In ACA-DCE analysis, rates of uninsurance decreased (DID -5.64, 95% confidence interval [CI] -7.23 to -4.04%, p<0.001) among patients aged 19–25 relative to older patients aged 26–64. There was no significant DID in advanced stage at diagnosis (stage≥II; p = 0.6) or orchiectomy more than 14 days after diagnosis (p = 0.6). For patients who received chemotherapy or radiotherapy as their first course of treatment, treatment greater than 60 days after diagnosis decreased (DID -4.84%, 95% CI -8.22 to -1.45%, p = 0.005) among patients aged 19–25 relative to patients aged 26–64. In Medicaid expansion states, rates of uninsurance decreased (DID -4.20%, 95% CI -7.67 to -0.73%, p = 0.018) while patients receiving chemotherapy or radiotherapy greater than 60 days after diagnosis decreased (DID -8.76, 95% CI -17.13 to -0.38%, p = 0.040) compared to rates in non-expansion states. No significant DIDs were seen for stage (p = 0.8) or time to orchiectomy (p = 0.1).

available from the National Cancer Database upon request and application by investigators by emailing the NCDB at NCDB_PUF@facs.org. The data for this analysis was obtained from the "Testis (Testis) special set(0->90+ )" Participant User File, application ID 2016.797.

**Funding:** This work was supported in part by the 2019 Urology Care Foundation Residency Research Award Program and the Russell Scott, Jr., MD Urology Research Fund (ABW). https://www.auanet.org/research/research-funding/aua-funding/residency-research-awards/residency-research-awards-fellows The funders had no role in study design, data collection and analysis, decision to publish, or preparation of the manuscript.

**Competing interests:** The authors have declared that no competing interests exist.

## Conclusions

Men with testicular cancer had lower uninsurance rates and decreased time to delivery of chemotherapy or radiotherapy following ACA-DCE and Medicaid expansions. Time to orchiectomy and stage at diagnosis did not change following either insurance expansion.

## Introduction

Testicular cancer is the most common cancer among adolescent and young adult men [1]. Young adults have historically also had the highest rates of no health insurance in the US [2]. Among many factors contributing to long-term outcomes, being under-insured has been linked to worse cancer outcomes, in particular among young men [3]. Uninsured men with testicular cancer are more likely to present at later stages of disease and have worse mortality outcomes [4]. Additionally, insurance coverage impacts type of treatment received for testicular cancer [5].

Recently, changes in the Affordable Care Act Dependent Care Expansion (ACA-DCE) have significantly increased coverage and access to care for young adults between the ages of 19 and 25 [6–8]. Once these changes took effect in 2010, young adults were allowed to remain covered under their parents' plans until the age of 26. Revisions to the ACA also allowed for several states to expand Medicaid eligibility to include U.S. citizens whose income falls below 133% of the federal poverty level and several did in January 2014. As of the 2017 fiscal year, over 12.6 million Americans were newly eligible and received coverage due to state expansion of Medicaid coverage [9]. Accordingly, rates of cancer patients without insurance decreased in states that expanded Medicaid relative to those in states that did not expand [3].

Previous works have assessed the associations between insurance expansion and outcomes specific to certain tumor types such as prostate and breast cancer [10, 11], which have shown differential benefits of insurance expansion for young patients with cancer depending on the cancer type. Thus, assessment of insurance expansion on outcomes for young patients with testicular cancer warrant investigations particularly since these patients present at young ages. To that end, we hypothesized the ACA-DCE in 2010 and widespread Medicaid expansion in 2014 impacted men presenting with testicular cancer by decreasing the percentage of those with no insurance and late stage disease ($\geq$II). We also hypothesized insurance expansion was associated with decreases in the percentage of men receiving delayed treatment for testicular cancer. We were able to answer these questions using a large national dataset in the US to compare each outcome before and after each expansion.

## Materials and methods

### Patients

Institutional Review Board exemption was granted for this study from Northwestern University (STU00210266). Data were obtained and accessed on May 6, 2019 and these data were fully anonymized before access was available. Because the data were without any patient identifiers, no consents were obtained. The National Cancer Database is a hospital-based cancer registry in the United States organized by the American Cancer Society and the American College of Surgeons [12]. It captures data on over 70% of all new cancer in the United States. For the analysis of the ACA-DCE, we included all male patients (n = 49,221, 100%) diagnosed with testicular cancer ages 19–64 from 2007 to 2009 and 2011 to 2016. Patients were excluded if they

had missing data on regional income or high school attainment (n = 666, 1.4%), insurance type (n = 959, 1.9%), or stage at diagnosis (n = 6,267, 12.7%). For the Medicaid expansion analysis, we included all male patients (n = 5,601, 100%) diagnosed with testicular cancer ages 40–64 from 2011 to 2013 and 2015 to 2016 residing in states that either expanded Medicaid on January 1, 2014 or never expanded Medicaid. In this analysis, there was only data available in the NCDB on Medicaid expansion for patients age 40 years or older. Patients were excluded if they had missing data on regional income or high school attainment (n = 71, 1.3%), insurance type (n = 88, 1.6%), or stage at diagnosis (n = 881, 15.7%).

## Independent variables

All independent variables were between-subjects. The main exposure of interest was year of diagnosis in both analyses: pre- (2007–2009) and post-ACA-DCE (2011–2013) and pre- (2011–2013) and post-Medicaid expansion (2015–2016). The years 2010 and 2014 were excluded as washout years for their respective expansions. Other covariates included in all adjusted regressions included race/ethnicity (Non-Hispanic White, non-Hispanic Black, Hispanic, or other/unknown), and Charlson/Deyo comorbidity index (0, 1, >1) [13]. Each regression was adjusted for zip code, median household income, and rate of adult high school attainment. These values were based on the 2012–2016 American Community Survey and were categorized based on quartiles relative to the entire United States. (www.census.gov/programs-surveys/acs)

## Outcomes

Our outcomes included proportion of patients without insurance coverage, proportion of patients with advanced stage at diagnosis (American Joint Committee on Cancer edition 7; ≥II [14]) in those with staging information, days from diagnosis to orchiectomy in patients whose first treatment was orchiectomy (<14 versus ≥14 days), and days from diagnosis to chemotherapy or radiotherapy in patients whose first treatment was either chemotherapy or radiotherapy (<60 versus ≥60 days). In the absence of specific guidelines on treatment timing, the cutoffs for timing of treatment were chosen *a priori* as generally acceptable timeframes as reflections of favorable (earlier treatment) versus unfavorable (delayed treatment) access to healthcare.

## Statistical analysis

A simple pre- and post-exposure comparison of our outcomes of interest would not account for any factors external to insurance expansion. It would also not account for trends that were already present prior to expansion. Thus, we performed difference-in-difference (DID) analyses based on the exposure to insurance expansion [15]. This method addresses the issues of external factors that could affect outcomes by using a comparison group that experiences the same external factors but does not experience the exposure (insurance expansion). In the analysis of the ACA-DCE, patients were considered to be exposed to the "intervention" if they were age 19–25 at the time of diagnosis. Controls were those aged 26–64 as this age group would not have been affected by the ACA-DCE and would have been too young to receive Medicare [8]. In the Medicaid expansion analysis, patients were considered to be in the "intervention" group if they resided in a state that expanded Medicaid on January 1, 2014. Controls were those patients who resided in states that never expanded Medicaid. Using multivariable linear regression for each outcome with an interaction term between the intervention/controls and pre- and post-exposure years of diagnosis, we calculated the DID of the percentage for each outcome to assess how each outcome in the exposed groups changed relative the non-

exposed groups before and after insurance expansion. A separate dummy variable was created for year of diagnosis for the pre-exposure years and the individual years following exposure for the ACA-DCE analysis (2015 alone and 2016 alone). This was not done for the Medicaid expansion analysis given the low numbers of patients. Subgroup analyses were performed limiting our analysis to patients living in zip code regions of low income (<$40,277 annual median household income). All statistical tests were performed using Stata 13 (College Station, TX) and p<0.05 was considered statistically significant.

## Results

### ACA-DCE analysis

**Patient characteristics.** In total, 8,026 patients age 19–25 years and 33,303 patients age 26–64 years were included in the final analysis (**S1 Table**). The median age at diagnosis for the patients age 19–25 years was 23 years vs 36 for patients age 26–64 years (p<0.001). In the 19–25 age group, fewer patients were white (70% vs 79%, p<0.001), more had zero comorbidities (95% vs 93%, p<0.001), and more resided in areas of low income and high non-high school attainment (both <0.001). Among those with information on treatment type and timing of treatment (n = 33,594), 18,996 (57%) received orchiectomy as their first treatment while 14,598 (43%) received chemotherapy or radiotherapy as their first form of treatment.

**Insurance.** Over the entire study period, 15% of patients age 19–25 years and 10% of those age 26–64 years had no insurance coverage (**S1 Table** and **Fig 1A**). The adjusted DID indicated rates of uninsurance decreased -5.64% (95% confidence interval [CI] -7.23 to -4.04%, p<0.001: **Table 1** and **S2 Table**) relative to older patients. Comparing the pre-ACA-DCE era (2007–2009) to the year 2016 revealed a DID of -7.43% (95% CI -10.12 to -4.73%, p = <0.001; **S3 Table**). When limiting the analysis to patients living in regions of low income, there was no statistically significant change in rates of uninsurance (p = 0.080).

**Stage at diagnosis.** In total, 36% of patients age 19–25 and 27% of those age 26–64 presented with advanced disease (Stage ≥II) at diagnosis (**S1 Table** and **Fig 1B**). No significant changes in advanced disease was seen among all patients (p = 0.6) or those living in low income areas (p = 0.9: **Table 1** and **S2 Table**).

**Days from diagnosis to orchiectomy.** Among patients who received orchiectomy as their first form of treatment, treatment was received 14 or more days after diagnosis for 9% of those age 19–25 and 10% of those age 26–64 (**S1 Table** and **Fig 1C**). No significant changes in days to orchiectomy was seen among all patients (p = 0.6) or those living in low income areas (p = 0.4: **Table 1** and **S2 Table**).

**Days from diagnosis to chemotherapy or radiotherapy.** Among patients who received chemotherapy or radiotherapy as their first form of treatment, treatment was received 60 or more days after diagnosis for 21% of patients age 19–25 and 21% of those age 26–64 (**S1 Table** and **Fig 1D**). Adjusted DID analysis showed a decrease in this figure of -4.84% (95% CI -8.22 to -1.45%, p = 0.005; **Table 1** and **S2 Table**). Comparing the pre-ACA-DCE era to the year 2016 alone showed an adjusted DID of -5.38% (95% CI -10.78 to 0.02%, p = 0.051; **S3 Table**). There was no change in time to chemotherapy or radiotherapy when the analysis was limited to patients living in regions of low income (p = 0.4).

### Medicaid expansion analysis

**Patient characteristics.** In total, 2,296 patients in Medicaid expansion states and 2,265 patients in non-expansion states were included in the final analysis (**S4 Table**). In the expansion group, more patients were white (88% vs 82%, p<0.001), and fewer resided in areas of low income and high non-high school attainment (both <0.001). Among those with

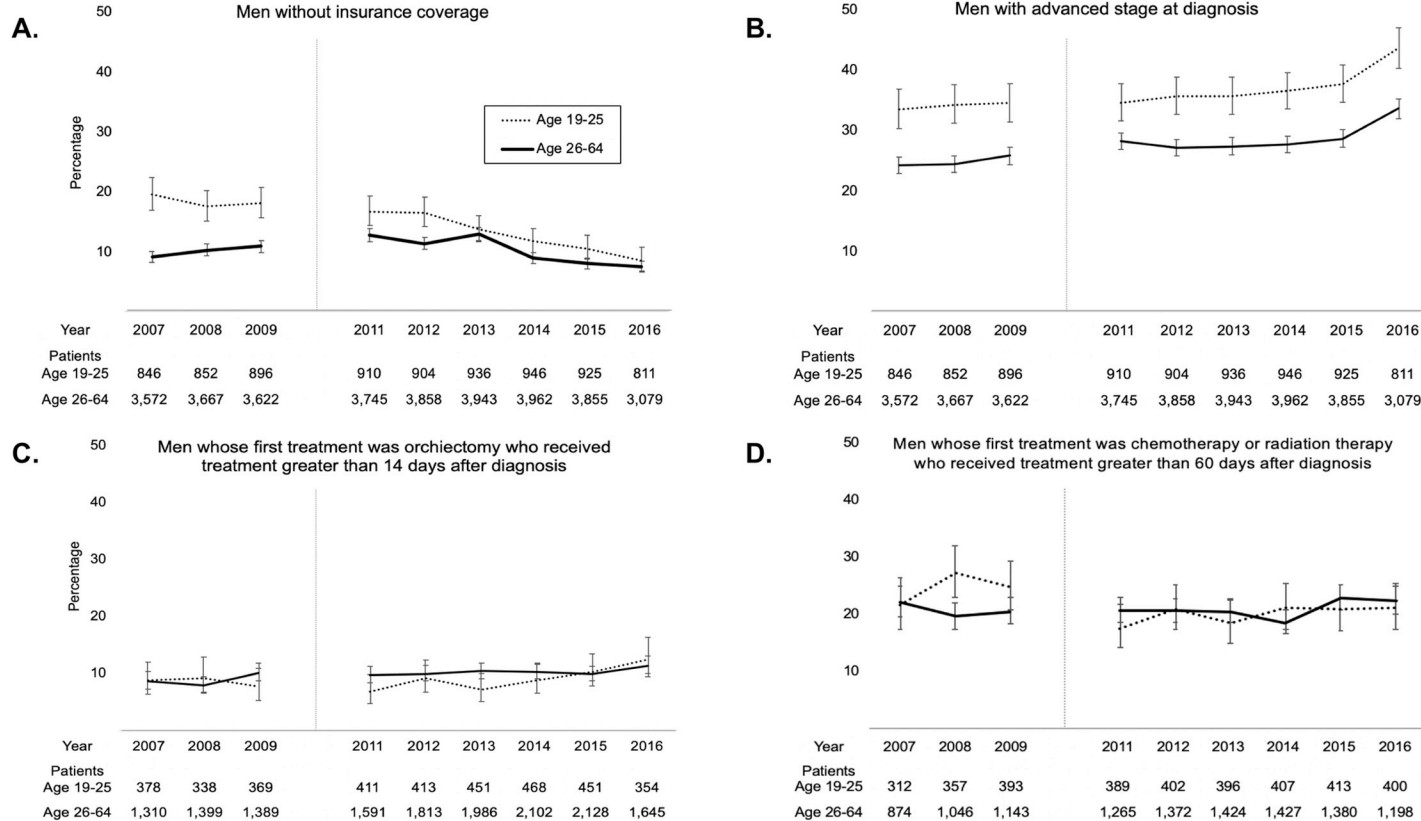

**Fig 1. Unadjusted temporal trends before and after ACA-DCE.** Unadjusted temporal trends comparing (a) patients without insurance coverage, (b) patients with advanced stage at diagnosis, (c) patients whose first treatment was orchiectomy who received treatment greater than 14 days after diagnosis, and (d) patients whose first treatment was chemotherapy or radiotherapy who received treatment greater than 60 days after diagnosis, between those who qualified for ACA-DCE coverage (ages 19–25) and those who didn't (ages 26–64), pre- (2007–2009) and post-ACA-DCE (2011–2016). Vertical lines demarcate initiation of the ACA-DCE. Abbreviation: ACA-DCE, Affordable Care Act Dependent Care Expansion.

information on treatment type and timing of treatment (n = 3,790), 2,217 (58%) received orchiectomy as their first treatment while 1,573 (42%) received chemotherapy or radiotherapy as their first form of treatment.

**Insurance.** Over the entire study period, 6% of patients in expansion states and 13% of those in non-expansion states had no insurance coverage (**S4 Table** and **Fig 2A**). Adjusted DID showed rates of uninsurance decreased -4.20% (95% CI -7.67 to -0.73%, p = 0.018; **Table 1** and **S5 Table**). When limiting the analysis to patients living in regions of low income, there was no change in rates of uninsurance (p = 0.055).

**Stage at diagnosis.** A total of 28% of patients in the expansion states and 31% of those in non-expansion states presented with advanced disease (Stage ≥II) at diagnosis (**S4 Table** and **Fig 2B**). No significant change in stage at diagnosis was seen among all patients (p = 0.8) or those living in low income areas (p = 0.4: **Table 1** and **S5 Table**).

**Days from diagnosis to orchiectomy.** Among patients who received orchiectomy as their first form of treatment, treatment was received 14 or more days after diagnosis for 13% of patients in expansion states and 11% of those in non-expansion states (**S4 Table** and **Fig 2C**). There was no significant change in overall time to orchiectomy following expansion (p = 0.1; **Table 1** and **S5 Table**). Adjusted DID analysis showed patients living in regions of low income were less likely to receive orchiectomy after 14 days following diagnosis (-23.35%, 95% CI -39.50 to -7.20%, p = 0.005).

**Table 1. Difference-in-difference analyses on outcomes for men with testicular cancer following ACA-DCE and Medicaid expansion.**

| Pre-expansion vs. post-expansion | All patients | | Regional low-income | |
|---|---|---|---|---|
| | Difference in difference (95% CI) | p | Difference in difference (95% CI) | p |
| % Patients without insurance coverage | | | | |
| ACA-DCE | -5.64 (-7.23 to -4.04) | **<0.001** | -4.64 (-9.84 to 0.56) | 0.080 |
| Medicaid Expansion | -4.20 (-7.67 to -0.73) | **0.018** | -11.80 (-23.85 to 0.24) | 0.055 |
| % Patients with advanced stage at diagnosis | | | | |
| ACA-DCE | -0.57 (-2.92 to 1.77) | 0.6 | -0.38 (-6.55 to 5.78) | 0.9 |
| Medicaid Expansion | -0.79 (-6.32 to 4.73) | 0.8 | -6.55 (-21.56 to 8.45) | 0.4 |
| % Patients whose first treatment was orchiectomy who received treatment greater than 14 days after diagnosis | | | | |
| ACA-DCE | -0.70 (-3.05 to 1.64) | 0.6 | 2.57 (-3.77 to 8.91) | 0.4 |
| Medicaid Expansion | -4.59 (-10.19 to 1.02) | 0.1 | -23.35 (-39.50 to -7.20) | **0.005** |
| % Patients whose first treatment was chemotherapy or radiotherapy who received treatment greater than 60 days after diagnosis | | | | |
| ACA-DCE | -4.84 (-8.22 to -1.45) | **0.005** | -3.80 (-12.35 to 4.75) | 0.4 |
| Medicaid Expansion | -8.76 (-17.13 to -0.38) | **0.040** | -14.74 (-36.02 to 6.53) | 0.2 |

Multivariable linear regression analyses were used to evaluate difference-in-differences for each outcome between intervention and controls, and pre- (2007–2009) and post-exposure years (2011–2016) for the ACA-DCE, and pre- (2011–2013) and post-exposure years (2015–2016) for Medicaid expansion. Covariates included in the adjusted analysis included patient age, race/ethnicity, Charlson-comorbidity index, regional income, and regional high school attainment. Bolded p values are statistically significant (p < 0.05); Abbreviation: ACA-DCE, Affordable Care Act Dependent Care Expansion; CI, confidence interval.

**Days from diagnosis to chemotherapy or radiotherapy.** Among patients who received chemotherapy or radiotherapy as their first form of treatment, treatment was received 60 or more days after diagnosis for 22% of patients in expansion states and 20% of those in non-expansion states. (**S4 Table** and **Fig 2D**). This figure decreased -8.76 (95% CI -17.13 to -0.38, p = 0.040) among all patients (**Table 1** and **S5 Table**) but did not change among those living in low income areas (p = 0.2).

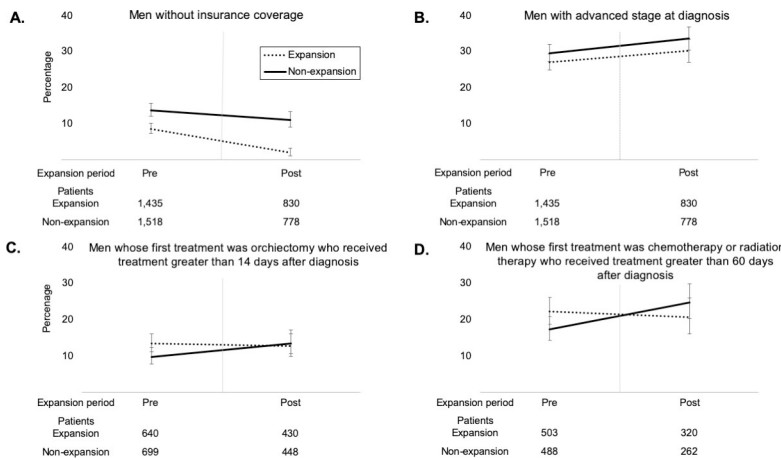

**Fig 2. Unadjusted temporal trends before and after Medicaid expansion.** Unadjusted temporal trends comparing (a) patients without insurance coverage, (b) patients with advanced stage at diagnosis, (c) patients whose first treatment was orchiectomy who received treatment greater than 14 days after diagnosis, and (d) patients whose first treatment was chemotherapy or radiotherapy who received treatment greater than 60 days after diagnosis, between those residing in Medicaid expansion and non-expansion states, pre- (2011–2013) and post-Medicaid expansion (2015–2016). Vertical lines demarcate when state-dependent participation in Medicaid Expansion began.

## Discussion

In this retrospective study, we used data from a national cancer registry to identify changes in the rate of insurance coverage, stage at diagnosis, and time to treatment in male patients diagnosed with testicular cancer following both the ACA-DCE and Medicaid expansion. Compared to the control group, patients age 19–25 and those in Medicaid expansion states experienced a significant decrease in rates of uninsurance following ACA-DCE in 2010 and Medicaid expansion in 2014, respectively. They both, likewise, experienced decreases in time to treatment for those whose first treatment was chemotherapy or radiotherapy relative to controls. There were no decreases noted in stage at diagnosis or time to treatment for those whose first treatment was orchiectomy.

The ACA-DCE and Medicaid expansions have prompted an examination of the effects of increased insurance coverage on cancer outcomes. There has been a significant decrease in the percentage of uninsured, non-elderly patients with newly diagnosed cancer following implementation of the ACA in both Medicaid expansion and non-expansion states, but with greater magnitude in Medicaid expansion states [3, 16]. A more recent study showed higher insurance rates did not result in changes in time to treatment for young women with breast cancer [10]. Notably, though, among patients with newly diagnosed prostate cancer, Medicaid expansion was associated with decreases in the proportion of patients presenting with high-risk disease [11]. Thus, recent insurance expansions may differentially impact patients with different cancer.

Because testicular cancer is ordinarily diagnosed at a relatively young age [1], insurance expansion via the ACA-DCE and Medicaid expansion had substantial potential to positively impact patients with newly diagnosed testicular cancer. According to recent reports, young men aged 20–34 have the highest incidence of being both uninsured and diagnosed with testicular cancer [17–19]. Additionally, previous work that analyzed data collected from the Surveillance, Epidemiology and End Reports database demonstrated lack of insurance in patients with testicular cancer increases risk of presenting with advanced stage disease, which is associated with worse mortality [4, 20]. This makes young men with testicular cancer a vulnerable population, yet likely to benefit from recent insurance expansions. Our analysis indicates decreases in the rates of uninsured patients with testicular cancer with following both ACA-DCE and Medicaid expansion. However, our findings also indicate neither policy change was associated with earlier staging at diagnosis. This is in contrast to prior studies indicating reductions in uninsured rates and late stage disease for all young patients with cancer following the ACA-DCE [16, 21, 22]. These inconsistencies may be in part due to relatively short term follow up data following each policy change. Longer-term follow-up may continue to show improvements in insurance coverage and outcomes for patients with testicular cancer including survival.

Notably, the rates of uninsurance or stage at diagnosis did not measurably change among men from regions of low annual income. However, the number of men in these analyses were small for both expansions (**S1 and S4 Tables**). Lack of data from patients age 19–40 in the Medicaid expansion dataset likely reduced the power of the Medicaid expansion analysis. Additionally, men from backgrounds of higher socioeconomic status and previous insurance coverage in childhood were more likely to take advantage of the ACA-DCE and parental coverage [23]. Further work is needed to understand the benefits of insurance expansion on men of low socioeconomic status.

In addition to staging, earlier treatment has also been linked to better survival and disease outcomes [24]. Prior studies have demonstrated insurance expansion is associated with improvements in access and timeliness of treatment for pancreatic, thyroid, and colorectal

cancer [25, 26]. Mirroring these results, our findings noted ACA-DCE and Medicaid expansion were associated with a decrease in time to treatment in men whose first treatment was either chemotherapy or radiotherapy. Additionally, men from regions of low annual income experienced decreases in time to treatment if their first treatment was orchiectomy. These results are encouraging and may reflect ongoing positive trends in more timely treatment for men with testicular cancer.

Several limitations should be taken into consideration when reviewing this study's findings. First, NCDB only includes data from Commission on Cancer-accredited US hospitals. Accredited hospitals are more likely to be larger and have more cancer-related services available to patients, thus results may not be generalizable to the overall population [27]. Second, only information for men age 40 years or older was available for the Medicaid expansion analysis. Given the highest incidence of testicular cancer occurs in men aged 20–44 with a median of 33 years old, this limits the generalizability of these findings to all young men diagnosed with testicular cancer [17]. Third, relatively short-term follow-up data in conjunction with inclusion of the years of implementation for each policy change may have biased the results in ways that are difficult to ascertain. There were likely inevitable logistical delays in transitioning in new policy, which may have delayed changes in patients' access to care. Fourth, our outcomes are all still proxies for more meaningful oncologic outcomes such as cancer-specific survival and morbidities related to advanced stage cancer diagnosis. Additionally, unbalanced age at diagnosis between in the comparison groups in the ACA-DCE analysis may create inherent bias given all patients age 19 to 25 years would have benefited from ACA-DCE, thus an older "control" group was required. Several unmeasured relevant variables not included in the NCDB that may have affected our outcomes warrant mentioning including individual-level measurements of socioeconomic status, smoking history and body-mass index among others. The current study was also limited by relatively small sample sizes and short follow-up leading to reduced patient numbers for subgroup analyses. For this reason, ongoing work should continue to evaluate the differential effects of insurance expansion for vulnerable patient groups based on race and ethnicity. Finally, we acknowledge a large percentage of patients who were described as receiving chemotherapy or radiotherapy as their first form of treatment. This finding likely reflects either a selection bias based on recorded treatments or timing to treatments in the NCDB. It may also reflect a lack of recording orchiectomy if, for instance, the orchiectomy was performed prior to presentation to the NCDB facility. Thus, some of the patients recorded as receiving chemotherapy or radiotherapy as their first treatment may have received those treatments as adjuvant treatments.

In summary, data from the NCDB revealed ACA-DCE and Medicaid expansion were associated with decreases in the percentage of uninsured men with testicular cancer. Despite this, no association was found between either expansion and the staging of disease at diagnosis. However, both expansions were associated with decreases in time to treatment for men whose first treatment was chemotherapy or radiotherapy. Moving forward, future research should attempt to identify additional variables affecting testicular cancer outcomes such as race and education. Longer-term follow up should continue to assess outcomes for men with testicular cancer related to insurance expansions. Additionally, examination of the impact of insurance expansion on survival outcomes would offer a more comprehensive analysis on oncologic outcomes.

## Conclusion

In a national cohort of men with testicular cancer, ACA-DCE and Medicaid expansion were each independently associated with decreases in rates of uninsurance, but no association with

stage at diagnosis was found. Both expansions were associated with improvements in time to treatment in patients whose first treatment was chemotherapy or radiotherapy. Further understanding of the impact insurance expansions have had on healthcare for men with testicular cancer warrants longer-term follow up, inclusion of more patients aged 19–40 for Medicaid expansion analysis, and addition of cancer-specific survival outcomes.

## Supporting information

**S1 Table. Patient characteristics for Affordable Care Act Dependent Care Expansion analysis.** [a] Comparisons based on Pearson's Chi-squared analyses for discrete covariates and Mann-Whitney U test for age.
(DOCX)

**S2 Table. Raw data for difference-in-difference analyses for ACA-DCE.** These data were used to generate Table 1 and Fig 1.
(DOCX)

**S3 Table. Difference-in-difference analyses on outcomes for men with testicular cancer following ACA-DCE.** Multivariable linear regression analyses were used to evaluate difference-in-differences for each outcome between intervention and controls, and pre- (2007–2009) and individual post-exposure years (2011–2016). Bolded p values are statistically significant (p < 0.05). Abbreviation: ACA-DCE, Affordable Care Act Dependent Care Expansion; CI, confidence interval.
(DOCX)

**S4 Table. Patient characteristics for Medicaid expansion analysis.** [a] Comparisons based on Pearson's Chi-squared analyses for discrete covariates and Mann-Whitney U test for age.
(DOCX)

**S5 Table. Raw data for difference-in-difference analyses for Medicaid expansion.** These data were used to generate Table 1 and Fig 2.
(DOCX)

## Acknowledgments

**Disclaimer:** The NCDB is a joint project of the Commission on Cancer of the American College of Surgeons and the American Cancer Society. The data used in the study are derived from a de-identified NCDB file. The American College of Surgeons and the Commission on Cancer have not verified and are not responsible for the analytic or statistical methodology employed, or the conclusions drawn from these data by the investigators.

## Author Contributions

**Conceptualization:** Adam B. Weiner, Stephen Jan, Ketan Jain-Poster, Oliver S. Ko, Shilajit D. Kundu.

**Data curation:** Adam B. Weiner.

**Formal analysis:** Adam B. Weiner, Oliver S. Ko.

**Funding acquisition:** Adam B. Weiner.

**Investigation:** Adam B. Weiner, Ketan Jain-Poster, Oliver S. Ko, Anuj S. Desai.

**Methodology:** Adam B. Weiner, Stephen Jan, Ketan Jain-Poster, Oliver S. Ko.

**Project administration:** Shilajit D. Kundu.

**Resources:** Shilajit D. Kundu.

**Supervision:** Anuj S. Desai, Shilajit D. Kundu.

**Validation:** Stephen Jan, Oliver S. Ko.

**Visualization:** Stephen Jan, Ketan Jain-Poster, Oliver S. Ko, Anuj S. Desai.

**Writing – original draft:** Adam B. Weiner, Stephen Jan, Ketan Jain-Poster, Oliver S. Ko, Anuj S. Desai, Shilajit D. Kundu.

**Writing – review & editing:** Adam B. Weiner, Stephen Jan, Ketan Jain-Poster, Oliver S. Ko, Anuj S. Desai, Shilajit D. Kundu.

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
