## [Decision Letter · Decision Letter 0]

14 Jul 2020

PONE-D-20-10591

Insurance coverage, stage at diagnosis, and time to treatment following dependent coverage and Medicaid expansion for men with testicular cancer.

PLOS ONE

Dear Dr. Kundu,

Thank you for submitting your manuscript to PLOS ONE. After careful consideration, we feel that it has merit but does not fully meet PLOS ONE’s publication criteria as it currently stands. Therefore, we invite you to submit a revised version of the manuscript that addresses the points raised during the review process.

ACADEMIC EDITOR:

Address the following comments in addition to the reviewers’ comments

Abstract

The n’s in the methods of the abstract are a bit confusing. For example, it is not clear if n=41,329 refers to only the post-population. Revise for clarityIn the results of the abstract, clarify when results are for all men and when they are comparing 19-25 and 26+Be consistent with the decimal places for the non-significant p-values in the abstract. I will suggest you change p=0.109 to p=0.1

Methods

Line 120: Move n=49221,100% to line 119 after “all men”Line 126: Move n=5601,100% to line 123 after “all men”Lines 126-127 somehow conflicts with line 124. Revise one of these sentences as you cannot include 19-64 years if the database has information for men aged 40+ onlyProvide a rationale for selecting 14 days and 60 days for orchiectomy and chemotherapy or radiation respectivelyProvide a rationale for comparing the two different age groups

Results

Lines 180-183: Clarify the comparison groups. If you are comparing 19-25 to 26-64 then it is obvious that they will be younger. You could instead indicate that “the median age for 19-24 group was …”Provide the percentage of men that received orchiectomy as first line of therapyProvide the pre and post percentages for each age group in Table 1 in addition to the DID or in a different table

Discussion

I will not refer to this pre and post study as a cohort study

We look forward to receiving your revised manuscript.

Kind regards,

Ernest K. Amankwah, PhD

Academic Editor

PLOS ONE

Journal Requirements:

2.  In the ethics statement in the manuscript and in the online submission form, please provide additional information about the patient records used in your retrospective study, including: a) whether all data were fully anonymized before you accessed them and b) the date range (month and year) during which patients' medical records were accessed."

3. To comply with PLOS ONE submission guidelines, in your Methods section, please provide additional information regarding your statistical analyses, including the specific statistical tests performed in your analysis. For more information on PLOS ONE's expectations for statistical reporting, please see https://journals.plos.org/plosone/s/submission-guidelines.#loc-statistical-reporting

Reviewers' comments:

Reviewer's Responses to Questions

**Comments to the Author**

1. Is the manuscript technically sound, and do the data support the conclusions?

Reviewer #1: Yes

Reviewer #2: Yes

2. Has the statistical analysis been performed appropriately and rigorously? 

Reviewer #1: Yes

Reviewer #2: Yes

3. Have the authors made all data underlying the findings in their manuscript fully available?

Reviewer #1: Yes

Reviewer #2: Yes

4. Is the manuscript presented in an intelligible fashion and written in standard English?

Reviewer #1: Yes

Reviewer #2: Yes

5. Review Comments to the Author

Reviewer #1: PONE-D-20-10591

Insurance coverage, stage at diagnosis, and time to treatment following dependent coverage and Medicaid expansion for men with testicular cancer

Summary: The authors conducted a retrospective cohort study that investigated changes in rates of insurance coverage, cancer stage at diagnosis, and time to treatment among men with testicular cancer following introduction of the Affordable Care Act’s Dependent Care Expansion (ACA-DCE) in 2010 and the Medicare expansion program in 2014. Data were obtained from the National Cancer Database (NCDB). Because the ACA-DCE permits dependent coverage for adults between the ages 19 to 25 years, participants were considered exposed to the “intervention” if they were between 19 to 25 years at the time of diagnosis, while those between the ages 26 to 64 years at diagnosis were used as controls. For the Medicaid expansion analysis, patients were considered exposed to the “intervention” if they resided in a state that expanded Medicaid on January 1, 2014, while those who resided in states that did not adopt the Medicare expansion coverage were used as controls. For statistical analysis, multivariable-adjusted linear regression was used. The results show that compared to the control group, ACA-DCE beneficiaries of age 19 to 25 and patients in Medicaid expansion states experienced significant increases in medical insurance rates following ACA-DCE in 2010 and Medicaid expansion in 2014. These two groups also experienced a decrease in time to treatment for those whose first-line treatment was chemotherapy or radiotherapy as compared with controls, but no differences were observed in stage at diagnosis or time to treatment for those whose first-line therapy was orchiectomy. Overall, this study adds to the literature on the impact of ACA-DCE and the Medicare expansion program on health outcomes. Below are few suggestions for improvement.

1. For the ACA-DCE analysis, patients in the control group are older than those in the control group. This raises the concern of whether the observed differences could have been driven by age and not necessary the introduction of ACA-DCE. I wonder why the authors didn’t choose a control group of similar age range as the intervention group. Could the authors have used testicular cancer patients between age 19 to 25 years who did not benefit from ACA-DCE as controls?

2. Abstract: Please make explicitly clear which patients belonged to the “intervention” group versus the comparison groups. It is quite confusing to understand until I read the entire manuscript.

3. Abstract: In the concluding statement, please indicate that two of the hypotheses tested produced null results; i.e., no association between ACA-DCE coverage and advanced state at diagnosis or orchiectomy.

4. Methods: Stratified analysis by race/ethnicity would be of great interest as it will show whether the findings are consistent across racial minority groups or not.

5. The repeated use of “men” in the results section presupposes that women may have been included. Since testicular cancer occurs only in men, I suggest stating briefly in the first paragraph of results section that only men were included in the study and then reduce the use of “men” throughout this section.

6. Study limitations: Please include residual confounding by poorly measured factors (e.g., neighborhood income level and neighborhood education level, as opposed to individual level data) and confounding by unmeasured factors, such as smoking history, BMI, etc.

Reviewer #2: Thank you for the opportunity to review this manuscript. The purpose of the paper was to assess the impact of ACA-DCE on adults aged 19-25 and Medicaid expansion on outcomes for men with testicular cancer. The authors found rates of uninsurance decreased among men aged 19-25 relative to older men, and in Medicaid expansion states, rates of uninsurance also decreased. The authors provided valuable information that is significant to oncology regarding the relationship between expanded coverage and outcomes in testicular cancer in adolescent and young males. Overall, the manuscript is well organized, and the authors' data and analyses fully supported their hypotheses and provided statistical measures that captured the exposure to insurance expansion. I recommend the manuscript for publication with the following minor recommendations:

Introduction – The authors provided appropriate evidence to support their claim for their study. However, in the introduction (line 103), it is mentioned that this study is the first to be studied in adolescent and young adult males. Are there any prior studies in other areas of oncology and young adults to boost the importance of expanded coverage and delayed treatment in this population?

Results – The results are nicely presented in tables and sub-headings. Line 198, figure 1 is not visible, and the figure in line 250 is not listed here.

Discussion - Overall, the authors provided substantial evidence regarding the context of previous literature in comparison to their results. However, I would like to see more focus in the second paragraph regarding the impact of ACA-DCE and Medicaid expansions on increased insurance coverage on cancer outcomes. Was there only one study regarding increased insurance coverage on cancer outcomes? If so, was there any additional information on why higher insurance rates did not change outcomes for the study presented?

6. PLOS authors have the option to publish the peer review history of their article (what does this mean?). If published, this will include your full peer review and any attached files.

Reviewer #1: **Yes: **Samuel O. Antwi

Reviewer #2: No

---

## [Decision Letter · Decision Letter 1]

25 Aug 2020

Insurance coverage, stage at diagnosis, and time to treatment following dependent coverage and Medicaid expansion for men with testicular cancer.

PONE-D-20-10591R1

Dear Dr. Kundu,

We’re pleased to inform you that your manuscript has been judged scientifically suitable for publication and will be formally accepted for publication once it meets all outstanding technical requirements.

Kind regards,

Ernest K. Amankwah, PhD

Academic Editor

PLOS ONE

Additional Editor Comments (optional):

Reviewers' comments:

Reviewer's Responses to Questions

**Comments to the Author**

1. If the authors have adequately addressed your comments raised in a previous round of review and you feel that this manuscript is now acceptable for publication, you may indicate that here to bypass the “Comments to the Author” section, enter your conflict of interest statement in the “Confidential to Editor” section, and submit your "Accept" recommendation.

Reviewer #1: All comments have been addressed

Reviewer #2: All comments have been addressed

2. Is the manuscript technically sound, and do the data support the conclusions?

Reviewer #1: Yes

Reviewer #2: (No Response)

3. Has the statistical analysis been performed appropriately and rigorously? 

Reviewer #1: Yes

Reviewer #2: (No Response)

4. Have the authors made all data underlying the findings in their manuscript fully available?

Reviewer #1: Yes

Reviewer #2: (No Response)

5. Is the manuscript presented in an intelligible fashion and written in standard English?

Reviewer #1: Yes

Reviewer #2: (No Response)

6. Review Comments to the Author

Reviewer #1: The authors have mostly addressed the concerns cited in the initial review. There is one concern related to differential age range between the intervention groups. As explained by the authors, it is an inherent limitation in the administrative data used for the analyses and they have discussed this as part of the study limitations.

Reviewer #2: (No Response)

7. PLOS authors have the option to publish the peer review history of their article (what does this mean?). If published, this will include your full peer review and any attached files.

Reviewer #1: No

Reviewer #2: No

---

## [Editor Report · Acceptance letter]

26 Aug 2020

PONE-D-20-10591R1 

Insurance coverage, stage at diagnosis, and time to treatment following dependent coverage and Medicaid expansion for men with testicular cancer. 

Dear Dr. Kundu:

I'm pleased to inform you that your manuscript has been deemed suitable for publication in PLOS ONE. Congratulations! Your manuscript is now with our production department. 

Kind regards, 

on behalf of

Dr. Ernest K. Amankwah 

Academic Editor

PLOS ONE